

# Surface Radiation Trends at North Slope of Alaska Influenced by Large-Scale Circulation and Atmospheric Rivers

Dan Lubin[1], Xun Zou[1], Johannes Mülmenstädt[2], Andrew Vogelmann[3], Maria Cadeddu[4], Damao Zhang[2]

[1]Scripps Institution of Oceanography, University of California San Diego, La Jolla CA, 93093, USA
[2]Pacific Northwest National Laboratory, Richland WA, 99354, USA
[3]Brookhaven National Laboratory, Upton NY, 11973, USA
[4]Argonne National Laboratory, Lemont IL, 60439, USA

*Correspondence to*: Dan Lubin (dlubin@ucsd.edu)

**Abstract.** Arctic amplification manifests as a pervasive warming trend emerging over the past century in near-surface air temperature throughout the Arctic that is double the globally averaged temperature increase throughout most of the year. It results from complex processes involving oceanic, atmospheric and terrestrial components which require detailed study to discern roles of the fundamental processes involved to improve predictions of the Arctic environment. We report on signals that are beginning to emerge, on a timescale predicted by recent satellite remote sensing studies, from the unique 25-year record of detailed surface-based radiometer measurements obtained by the US Department of Energy Atmospheric Radiation Measurement (ARM) Facility North Slope of Alaska (NSA) site at Utqiaġvik, Alaska. Statistically significant warming trends are found at the site in the boreal fall, while a decrease in net radiation occurs in late summer. This decrease is driven primarily by the decrease in shortwave radiation resulting from increasing cloud liquid water path as observed by the microwave radiometer. Analysis of prevailing meteorological regimes linking NSA with the Arctic Ocean and subarctic latitudes, and atmospheric rivers, suggests that specific changing circulation patterns are the primary driver for these trends.

## 1 Introduction

Arctic amplification is recognized as a significant feature of the global climate system (Smith et al., 2014; Miller and Carter, 2015; Serreze and Barry, 2011). Its essence is a pervasive warming of near-surface air temperature throughout the Arctic over the past century, with warming since the start of the present century emerging as double the globally averaged temperature increase throughout most of the year although smaller during summer (Wendisch et al., 2023). Arctic amplification is a complex process involving a wide variety of oceanic, atmospheric and terrestrial components. Some, including sea ice and ice sheet retreat, involve the long-recognized ice-albedo feedback. Others involve more recently studied land and biosphere changes, aerosol influences on both the surface ice and snow cover and cloud microphysics, and dynamical interaction with subarctic latitudes (Wendisch et al., 2023). Atmospheric components also include effects such as lapse rate and cloud optical depth feedbacks (Taylor et al., 2022). While Arctic amplification emerges consistently in Earth



system model (ESM) simulations, large differences still exist within the simulated components requiring state-of-the-art observations to improve ESM physics and chemistry (e.g., Shupe et al., 2022).

Satellite remote sensing has played a vital role in the study of Arctic amplification (Esau et al., 2023). To date the most substantial remote sensing contributions have involved surface properties including trends in sea ice and ice sheet extent, and changes in land cover, while there has also been recent progress with satellite mapping of tropospheric aerosols (Swain et al., 2024). With respect to cloud properties and surface radiative fluxes, satellite remote sensing offers potential in the Arctic particularly over sea ice (Cesana et al., 2024), and related satellite remote sensing products have substantially informed Arctic climate model simulations (Tan and Storelvmo, 2019; Taylor et al., 2024; Tan et al., 2025) but significant retrieval uncertainties persist (e.g., Riihelä et al., 2017). Arctic surface and aircraft-based radiometric observations are an important resource for validating and refining satellite remote sensing retrievals (Smith et al., 2017; Di Biagio et al., 2020; Barrientos-Velasco et al., 2022), and, equally importantly, for detailed understanding of fundamental processes (Tjernström et al., 2014; Miller et al. 2017, Wendisch et al., 2024)**.**

A number of satellite remote sensing studies have focused on surface radiative fluxes and their governing cloud properties, particularly over the Arctic Ocean and during spring and autumn sea ice transitions (Wang and Key, 2005; Liu et al., 2008; Wang et al., 2012; Comiso and Hall, 2014; Huang et al., 2017; Sledd and L'Ecuyer, 2021; Wang et al., 2021; Lelli et al., 2023). With the larger uncertainties in satellite-based estimates of surface shortwave and longwave fluxes, as compared with the direct observations in this work, two of these studies have determined radiative forcing trends over 20-year periods that are not yet statistically significant, and have therefore reported a trend time of emergence (ToE) to 95% statistical confidence (Sledd and L'Ecuyer, 2021; Lelli et al., 2023). Here we will show in situ observation trends emerging at 95% confidence that are consistent with these satellite remote sensing projections.

One of the most advanced and comprehensive atmospheric observatories in the high Arctic is operated by the US Department of Energy (DOE) at Utqiaġvik, Alaska (71.323ºN, 156.615ºW), where the DOE Atmospheric Radiation Measurement (ARM) Facility maintains its North Slope of Alaska (NSA) Central Facility (Verlinde et al., 2016). Since the late 1990s the NSA suite of radiometric, cloud radar and lidar, and meteorological equipment has provided valuable climatological information about the region's cloud properties including seasonal variability (Dong and Mace, 2003) and comparison with other high latitude sites such as the Antarctic (Zhang et al., 2019; Desai et al., 2024). One important discovery from the modern instrumental era involving Arctic stratiform clouds is the persistence of radiatively significant liquid water content in all seasons and to temperatures even below 240 K (Intrieri et al., 2002; Shupe et al., 2013). Analysis of NSA data for mixed-phase clouds (Dong and Mace, 2003; Zhang et al., 2019) shows that cloud liquid water path (LWP) values as large as 60 g m$^{-2}$ are frequently observed during winter, and values > 100 g m$^{-2}$ are frequently observed throughout the sunlit part of the year, with largest values during spring and autumn. For attenuation of shortwave (SW) radiation at the





surface, liquid water content is the dominant component of the cloud optical depth in Arctic stratiform clouds (Lubin and Vogelmann, 2006). For this work NSA provides a nearly continuous time series of broadband SW and LW upwelling and downwelling surface radiation starting in 1999 (Michalsky and Long, 2016), and a similar time series of microwave radiometer (MWR) retrievals of cloud LWP and atmospheric precipitable water vapor (PWV) content starting in 2001 (Cadeddu et al., 2013). These data sets' consistency and quality control enables multidecadal trends in surface net radiative

fluxes to now emerge with statistical significance, particularly during summer. Some of these trends are consistent with similarly emerging trends in MWR-measured PWV and LWP. Combining these NSA observations with ERA5 meteorological reanalysis data (Hersbach et al., 2020), we show that these radiation and cloud property trends result mainly from varying moisture transport between subarctic and high Arctic latitudes, as opposed to local feedback between the surface temperature and the column water vapor and cloud optical depth.

**2 Data and Methods**

Meteorological data analysis is based on ERA5 (Hersbach et al., 2020). All radiometric quantities and cloud properties are from datasets in the DOE ARM Facility archive. Near-surface (2 m) air temperatures are obtained from ERA5 in the single grid cell containing the NSA site, because the ARM NSA meteorological datasets have many gaps during the earlier years. These 2 m air temperatures are diurnally averaged. To avoid the possibility of autocorrelation these diurnal averages, and

also those from the radiative flux and MWR-derived quantities, are averaged over the semi-monthly intervals so that there is only one data point per year in the time series analysed for trends. To better identify statistically significant trends and their potential physical causes we subdivide the year into these 24 semi-monthly intervals to account for the large seasonal cycles in both temperature and insolation. In this work trends are evaluated first using ordinary linear least squares (OLS), and second using a nonparametric Mann-Kendall (MK) test.


The upwelling and downwelling SW and LW radiative fluxes are obtained from the RADFLUX1LONG data product from September 2003 through November 2024. Between April 1999 and August 2003 the radiative fluxes are taken from the SKYRAD (downwelling) and GNDRAD (upwelling) data products. The measurement uncertainty in the ARM Facility broadband radiometers is relatively small, of order 5-10 W m$^{-2}$ (Bush et al., 2000; Wang and Dickinson, 2013). During the

overlap period between these products in September 2003 the discrepancies between the two are at least one order of magnitude smaller than a pyranometer's standard measurement uncertainty (e.g., Bush et al., 2000) and are therefore negligible. From these one-minute data the net (downwelling minus upwelling) fluxes are evaluated and averaged over each hour. PWV and LWP retrievals are obtained from the MWRRET data product. Before making hourly averages from the 30-second data, values of LWP > 500 g m$^{-2}$ are omitted as physically unrealistic and likely resulting from instrumental problems

such as transient riming on the microwave antenna. In addition to natural variability, these quantities are expected to have larger uncertainty than the radiative fluxes due to their determination by a remote sensing algorithm as opposed to direct



measurement. From the MWR typical PWV retrieval uncertainties are 5% (Cadeddu et al., 2013) while LWP retrievals from the NSA MWR may be in the range 25-30 g m$^{-2}$ (Turner et al., 2007); the latter being a large uncertainty compared with the climatological range in Arctic cloud LWP (Dong and Mace, 2003; Zhang et al., 2009).


The hourly averages of SW and LW fluxes, and PWV and LWP retrievals, are diurnally averaged then averaged over the semi-monthly interval if there are at least five days of data in the interval. Intervals with fewer than five days are omitted from the time series analysis. For the analysis involving single $k$-means clusters, these subsets often have fewer than five days of data in a semi-monthly interval. To maintain a useful sample size with these subsets while also minimizing the

possibility of autocorrelation, adjacent days of data within an interval are averaged into one data point while isolated days are considered by themselves. The resulting variable sample sizes for these subsets are indicated in the figures showing $k$-means clustering results.

We employ $k$-means clustering analysis for meteorological regime identification following Mülmenstädt et al. (2012) using

the fields: (1) 2 m air temperature anomaly, (2) surface pressure, (3) 2 m relative humidity anomaly, (4) 10 m zonal wind component, and (5) 10 meridional wind component. In this work the cluster centroids are evaluated from the ARM surface meteorology data (NSAMET) between 2004-2021. Anomalies are used for 2 m air temperature and relative humidity to account for the large seasonal variability. The anomalies are calculated relative to the 18-year mean for each month. To omit suspect data we let the ARM data ordering tool omit quality control-flagged data. Analysis was done using days when all

five fields have no quality control flags. The $k$-means clustering was redone with this longer NSAMET dataset, and a confusion matrix between the new and old (years 2004-2010) cluster classification results shows a constinency of >90%. The resulting cluster centroids $\mu_{jk}$ and their normalizations ($\sigma_j$) are shown in Table 1. To make the daily cluster identification in this work the five fields were taken from ERA5. The cluster to which a daily data point $x_j$ is associated is the one with minimum Euclidean distance:

$$d_k = \sqrt{\sum_{j=1}^{5}\left(\frac{x_j - \mu_{jk}}{\sigma_j}\right)^2},$$    (1)

where $d_k$ is evaluated for each of $k$ clusters using $j$ fields from Table 1.

Atmospheric rivers are detected over the NSA site following the polar-adapted AR scale (Zhang et al., 2024). Using ERA5 data with six-hourly time resolution in a 1° x 1° grid cell over the NSA site, the IVT is evaluated from the surface to the top

of the atmosphere. IVT is calculated as follows:

$$\text{IVT} = \sqrt{\left(\frac{1}{g}\int_{1000}^{10}\mathrm{q}u\mathrm{d}p\right)^2 + \left(\frac{1}{g}\int_{1000}^{10}\mathrm{q}v\mathrm{d}p\right)^2},$$    (2)





**Table 1. Cluster centroids $\mu_{jk}$ and normalizations $\sigma_j$ used to identify the specific meteorological regime in each day at NSA with ERA5 data.**

| Cluster | $T_{2m}$ (K) | $P_{sfc}$ (hPa) | $RH_{2m}$ (%) | $u_{10m}$ (m s$^{-1}$) | $v_{10m}$ (m s$^{-1}$) |
|---------|-----------|--------------|-----------|----------------------|----------------------|
| 1 | -4.78 | 1025.02 | -4.07 | -0.89 | -2.15 |
| 2 | -0.66 | 1008.94 | -2.89 | +0.17 | +2.39 |
| 3 | +0.35 | 1011.53 | +1.22 | -2.36 | -5.67 |
| 4 | +5.26 | 1015.05 | +5.34 | +1.56 | -0.98 |
| $\sigma_j$ | 5.25 | 10.13 | 5.55 | 2.79 | 4.36 |

where g is the gravity acceleration constant (m s$^{-2}$), q is specific humidity (kg kg$^{-1}$), u and v are zonal and meridional wind (m s$^{-1}$), and d$p$ is the differential pressure (hPa).If IVT > 100 kg m$^{-1}$ s$^{-1}$ for 24 h this is flagged as a Polar AR level 1 (AR P1). Durations exceeding 48 h then up to 72 h, or IVT increasing to 150-200 then 200-250 kg m$^{-1}$ s$^{-1}$, promote the precursor AR to levels AR P2 and AR P3. Further increases in IVT into the range 250-1500 kg m$^{-1}$ s$^{-1}$ promote the event into the intensity strengths AR 1-5 with thresholds 250, 500, 750, 1000, 1250 and 1500 kg m$^{-1}$ s$^{-1}$, respectively. These five thresholds

are the same as in the global AR scale (Ralph et al., 2019). The three lower Polar AR thresholds are introduced to recognize that smaller IVT values transported in an AR frontal pattern can be associated with substantial impacts on a local cryosphere environment such as anomalous precipitation or surface melt.

## 3 Results

### 3.1 Observed Trends in Radiative Fluxes and Cloud Properties

For orientation we first examine the trends in the near-surface (2 m) air temperature at NSA. Trends for the full ERA5 time series (1959-2024) and the period encompassing the ARM NSA Facility data (1999-2024) are given in Figure 1 and Table 2. Over the full time series 20 of 24 intervals (83%) show statistically significant upward trends, signifying that Arctic amplification has consistently warmed the NSA region beyond the global-mean warming trend of 0.2ºC per decade, over recent multidecadal timescales. Over the most recent 25 years there is less consistent statistical significance, and some colder

intervals (FEB-Early, APR-Early and JUN-Early) show negative trends. The largest positive trends over the most recent 25 years occur during autumn and early winter, while summer positive trends are small and not statistically significant. This contrast in trend significance between the shorter and longer time series suggests that various components of Arctic amplification may be operating over NSA during different seasons and may also be time-varying, or that some trends may need more than 25 years to become statistically significant.


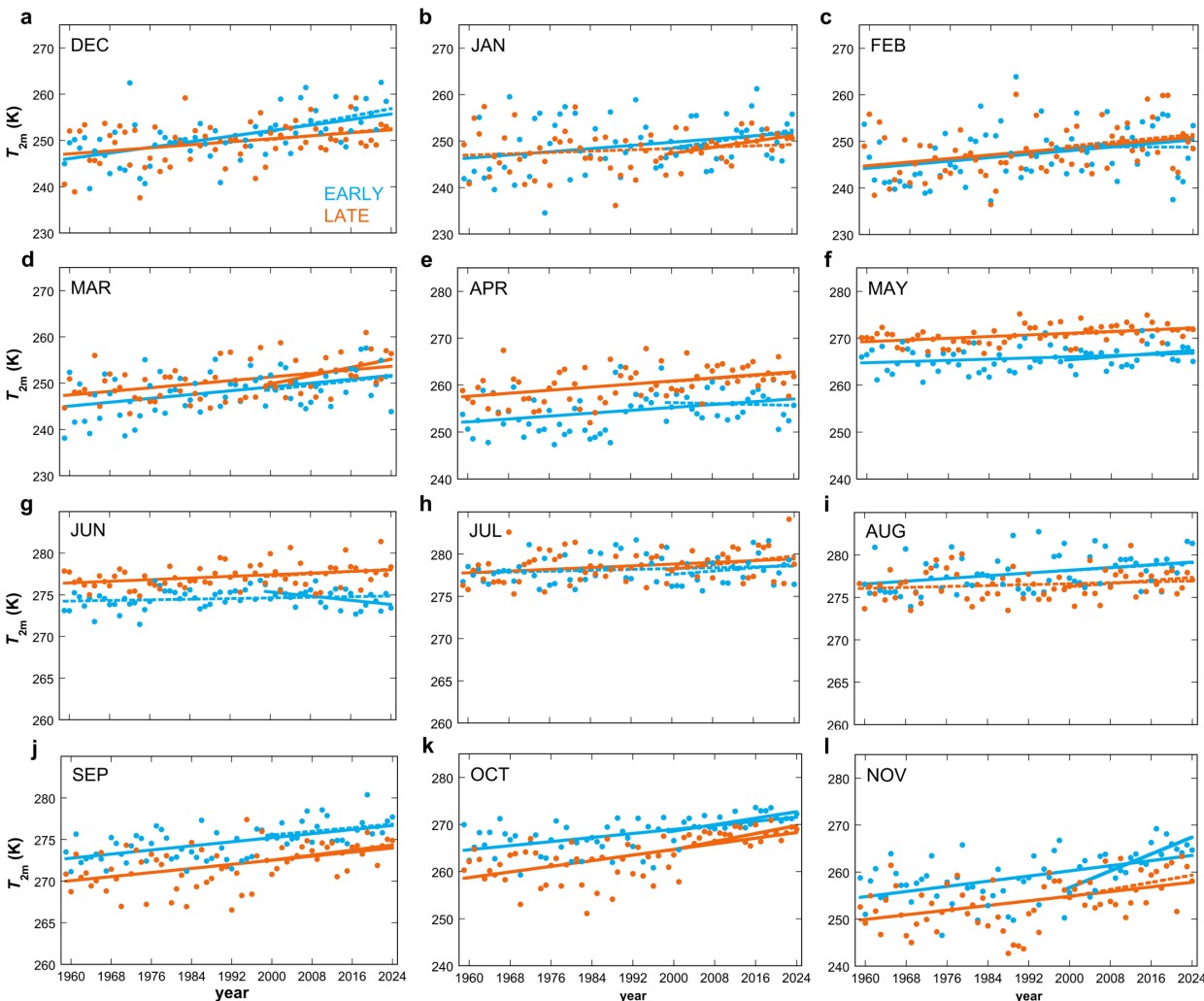

**Figure 1. Trends in the semi-monthly interval-averaged 2 m near-surface air temperature at the NSA site from ERA5 data, organized by month (panels a-l) and semi-monthly intervals (light blue for the first two weeks, orange for the second). For each interval two linear trend lines are shown, one for the full time series (1959-2024) and one for the period overlapping the NSA data (1999-2024). Trends that are statistically significant with two-tailed confidence are shown as solid lines, while trends with less statistical significance are shown as dotted lines.**

The trends in net radiative fluxes in the intervals with the most statistical significance are shown in Figure 2, and the full set of net radiative flux trends is shown in Table 3. Given the measurement uncertainty in the ARM Facility broadband radiometers (of order 5-10 W m$^{-2}$), the variability shown throughout Figure 2 reflects mainly the natural variability in both cloud amount and optical depth. The most consistent statistical significance in the trends occurs during July and August, during which each of the intervals shows significance in two of the three quantities, net shortwave (SW), net longwave (LW)





**Table 2. Summary of ERA5 surface temperature trends at the NSA site. Shown for each timespan are the linear temperature change over the timespan and the Pearson correlation coefficient. Two-tailed (one-tailed) significant correlations are indicated in bold (italic).**

| | Whole Time Series | | 1959-1998 | | 1999-2024 | |
|---|---|---|---|---|---|---|
| Interval | $d_{59-24}$(K) | $r$ | $d_{59-98}$(K) | $r$ | $d_{99-24}$(K) | $r$ |
| JAN Early | **5.6** | **0.2899** | 5.0 | 0.2385 | 4.2 | 0.2665 |
| JAN Late | 2.4 | 0.1614 | -2.3 | -0.1402 | **4.2** | **0.4119** |
| FEB Early | **6.2** | **0.3141** | **6.9** | **0.3462** | -0.1 | -0.0038 |
| FEB Late | **6.3** | **0.3457** | 0.7 | 0.0389 | 2.6 | 0.1528 |
| MAR Early | **6.8** | **0.4422** | **6.6** | **0.4245** | 3.2 | 0.2378 |
| MAR Late | **6.4** | **0.4652** | 2.8 | 0.2243 | **5.8** | **0.4372** |
| APR Early | **5.1** | **0.3608** | **5.0** | **0.3251** | -0.6 | -0.0643 |
| APR Late | **5.4** | **0.4281** | **4.3** | **0.3236** | 2.0 | 0.2112 |
| MAY Early | **2.1** | **0.2491** | 1.1 | 0.1211 | 2.1 | 0.2810 |
| MAY Late | **3.0** | **0.4419** | **2.1** | **0.3227** | 1.4 | 0.2317 |
| JUN Early | 0.6 | 0.1577 | **1.9** | **0.4547** | **-1.6** | **-0.4413** |
| JUN Late | **1.6** | **0.3373** | **1.2** | **0.3219** | 0.9 | 0.1562 |
| JUL Early | 0.8 | 0.1536 | **1.9** | **0.3573** | 1.3 | 0.2201 |
| JUL Late | **1.7** | **0.2725** | 1.5 | 0.2558 | 1.7 | 0.2696 |
| AUG Early | **2.7** | **0.3646** | *2.0* | *0.2703* | 1.0 | 0.1560 |
| AUG Late | 0.9 | 0.1566 | 0.1 | 0.0090 | 1.2 | 0.2158 |
| SEP Early | **4.1** | **0.5894** | 0.9 | 0.1591 | 1.4 | 0.2660 |
| SEP Late | **4.1** | **0.4822** | 1.2 | 0.1338 | **2.0** | **0.4174** |
| OCT Early | **7.1** | **0.5961** | 1.2 | 0.1118 | **4.4** | **0.6326** |
| OCT Late | **9.9** | **0.6218** | 0.8 | 0.0608 | **5.5** | **0.5798** |
| NOV Early | **9.1** | **0.5283** | 2.1 | 0.1354 | **11.6** | **0.7751** |
| NOV Late | **8.2** | **0.4592** | -0.1 | -0.0066 | *5.0* | *0.3749* |
| DEC Early | **6.0** | **0.5587** | 2.9 | 0.1963 | *5.4* | *0.3421* |
| DEC Late | **5.4** | **0.3478** | 2.8 | 0.1750 | 2.6 | 0.2185 |




Figure 2. Time series of the semi-monthly interval-average net radiative flux components (shortwave SW and longwave LW) and their sum (Net); for the four intervals in July and August (panels a-l), and for net radiative flux in three other intervals that show some degree of statistical significance in the trend from 1999-2024 (panels m-o). The mean value in each interval is shown as a dot and the error bars are plus and minus one standard deviation about the mean. The linear least-squares trend is shown as a dotted line. The three numbers in each panel are the linear change in the radiative flux between 1999-2024 (top), the Pearson correlation coefficient from ordinary least squares (middle) and the percent confidence level in trend detection from a Mann-Kendall test (bottom). For the latter two numbers, values having statistical significance of 95% or greater are shown in blue, values with one-tailed confidence (90-95%) are shown in orange, and values with less statistical confidence are left in black.





**Table 3. Summary of the NSA surface radiation trends. *N* is the number of years having available data in each semi-monthly interval. Shown for each net radiative flux component are the linear change from 1999-2024, the Pearson correlation coefficient and the percent confidence in trend detection from a Mann-Kendall test. Two-tailed (one-tailed) significant correlations are indicated in bold (italic).**

| Net Fluxes (W m$^{-2}$) | | Shortwave | | | Longwave | | | Net Radiation | | |
|---|---|---|---|---|---|---|---|---|---|---|
| Interval | $N$ | $d_{99\text{-}24}$ | $r$ | M-K (%) | $d_{99\text{-}24}$ | $r$ | M-K (%) | $d_{99\text{-}24}$ | $r$ | M-K (%) |
| JAN Early | 24 | | | | -2.0 | -0.0744 | 61 | -1.9 | -0.0704 | 61 |
| JAN Late | 24 | | | | 3.2 | 0.0983 | 61 | 4.2 | 0.1285 | 66 |
| FEB Early | 24 | | | | -0.5 | -0.0155 | 59 | -0.1 | -0.0036 | 57 |
| FEB Late | 24 | | | | 7.8 | 0.3258 | 89 | **8.2** | 0.3370 | **95** |
| MAR Early | 23 | 0.6 | 0.0587 | 50 | **15.4** | **0.5067** | 99 | **16.0** | **0.6158** | **>99** |
| MAR Late | 25 | 1.8 | 0.1630 | 85 | 5.6 | 0.2490 | 88 | 7.4 | *0.3836* | **96** |
| APR Early | 26 | 3.1 | 0.1861 | 84 | -4.6 | -0.1836 | 81 | -1.5 | -0.0715 | 59 |
| APR Late | 26 | 1.9 | 0.0850 | 66 | -3.3 | -0.1015 | 68 | -1.4 | -0.0560 | 54 |
| MAY Early | 26 | -7.1 | -0.2786 | 93 | **11.9** | **0.4670** | **98** | 4.8 | 0.1940 | 92 |
| MAY Late | 26 | -0.9 | -0.0126 | 69 | -1.1 | -0.0396 | 52 | -2.1 | -0.0331 | 72 |
| JUN Early | 26 | -13.9 | -0.0964 | 57 | 4.0 | 0.1009 | 59 | -9.7 | -0.0852 | 54 |
| JUN Late | 26 | -8.3 | -0.1235 | 67 | 5.3 | 0.2050 | 82 | -3.0 | -0.0647 | 54 |
| JUL Early | 26 | **-28.0** | *-0.3524* | **98** | **12.5** | *0.3561* | **97** | **-15.5** | -0.3164 | **97** |
| JUL Late | 26 | **-45.9** | **-0.5001** | **99** | **19.7** | **0.5273** | **>99** | **-26.3** | **-0.4541** | **99** |
| AUG Early | 26 | **-16.8** | **-0.4061** | 93 | 4.9 | 0.2427 | 90 | **-11.9** | **-0.4112** | **97** |
| AUG Late | 26 | **-15.1** | **-0.4605** | **99** | 5.1 | 0.2012 | 89 | **-10.0** | **-0.4529** | **97** |
| SEP Early | 23 | -1.8 | -0.0775 | 70 | 5.0 | 0.2452 | 92 | 3.2 | 0.1662 | 89 |
| SEP Late | 25 | 0.8 | 0.0234 | 64 | 0.0 | 0.0007 | 60 | -4.5 | -0.1273 | 51 |
| OCT Early | 24 | 2.3 | 0.1365 | 65 | -2.0 | -0.0708 | 53 | 0.3 | 0.0129 | 55 |
| OCT Late | 24 | -0.7 | -0.1734 | 81 | 2.2 | 0.0924 | 51 | 1.4 | 0.0697 | 55 |
| NOV Early | 24 | | | | 4.1 | 0.1845 | 81 | 4.2 | 0.1958 | 79 |
| NOV Late | 24 | | | | 10.5 | 0.3171 | 92 | 10.9 | 0.3307 | 93 |
| DEC Early | 23 | | | | 1.5 | 0.0528 | 54 | 1.6 | 0.0543 | 54 |
| DEC Late | 23 | | | | 4.4 | 0.1498 | 70 | 4.4 | 0.1498 | 70 |



or their sum (Net). All four of these intervals show a decreasing SW trend with some degree of statistical significance. The LW trend is upward in all of these intervals, statistically significant in July but not in August. As the SW decreases outpace the LW increases the Net radiative flux in all four intervals shows a statistically significant downward trend. This is evidence of a gradual radiative surface cooling between 1999-2024, which might partially explain why the concomitant summer 2 m air temperature trends (Figure 1) are small and not statistically significant.

We see statistically significant Net radiative flux increases for three colder intervals (FEB-Late, MAR-Early and MAR-Late), also shown in Figure 2m-o. During most of the autumn and winter intervals there are hints of increasing LW and Net fluxes (Table 3) but as analysed here they do not yet rise to a level of statistical significance. In addition we notice that during the JUN-Early interval the magnitudes of the SW and Net flux trends are almost as large as those of the significant July and August trends, but their statistical significance is poor (Table 3). This is due to the wide variability in surface albedo during this interval, which is a transition period in snow cover. Outside this interval the surface albedo is consistently less than 0.2 during warmer months and greater than 0.7 during colder months. Figure 3 shows that this interval exhibits an obvious correlation between surface albedo and SW net flux, along with wide variability in both these quantities. The same is true during the opposite surface transition interval SEP-Late.

To explain the trends in summertime radiative fluxes, we examine the MWR retrievals of PWV and LWP. In addition to natural variability, these quantities are expected to have larger uncertainty than the radiative fluxes due to their determination by a remote sensing algorithm as opposed to direct measurement. Statistically significant trends in MWR-retrieved quantities thus appear in fewer intervals than for the radiative flux measurements, but some trends emerge during summer that correspond to the radiative flux trends (Figure 4, Table 4). Increasing PWV trends appear in all JUN-Late and all July and August intervals, and appear statistically significant during July corresponding to the statistically significant upward LW flux trends (Figure 2b,e). This is consistent with a steadily warming lower troposphere. LWP shows no trend in JUL-Early or AUG-Early, but shows statistically significant increasing trends in JUL-Late and AUG-Late. Thus the summer SW and Net flux trends are not yet entirely explained by the present MWR data time series, but a relationship with increasing LWP is emerging for two intervals. We also see statistically significant upward LWP trends during two spring intervals (Figure 4i,j) that correspond to significant upward LW flux trends (Table 3), and also a significant upward Net flux trend in MAR-Early (Figure 2n). A significant upward LWP trend in NOV-Early is associated with increasing LW and Net fluxes but these are not yet showing as significant (Table 3). The same is true for a significant upward PWV in JAN-Late (Figure 4l).

### 3.2 Synoptic-Scale Meteorological Influences

The NSA region is influenced by climatologically persistent cyclonic activity in the northwestern Pacific Ocean and anticyclones in the Beaufort/Chukchi Seas (Serreze et al., 1993; Serreze and Barry, 2014). Analysis of NSA meteorological





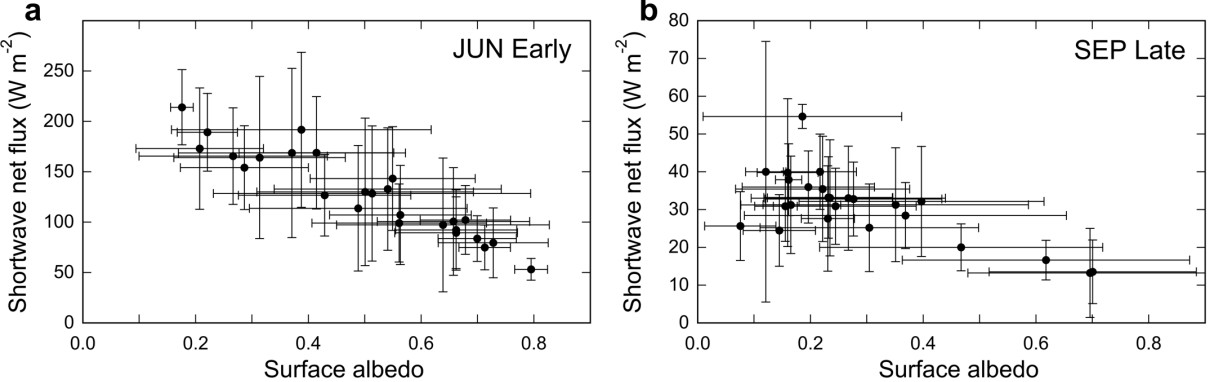

**Figure 3. Shortwave net flux versus surface albedo during the transitional semi-monthly intervals (a) early June and (b) late September. Error bars on both quantities are one standard deviation in the observations.**

data using *k*-means clustering (Mülmenstädt et al., 2012) has shown that these influences lead to four meteorological regimes (clusters) that manifest in all seasons. Cluster 1 describes the coldest regime dominated by the Beaufort Sea High (Serreze and Barry, 2014) and features mainly easterly to northeasterly near-surface winds at NSA. Under Cluster 1 conditions at NSA are closest to a "polar desert." Cluster 2 describes the second coldest regime, characterized by strong low pressure in the Arctic Ocean and Beaufort Sea connected to low pressure of average strength in the Gulf of Alaska. As the cyclonic flow will carry moisture to NSA from over the open Arctic Ocean even in the colder conditions, cloud cover under Cluster 2 is more extensive than under Cluster 1 but is limited in vertical extent. Cluster 3 describes a mixture of weak high pressure over the Arctic Ocean and strong low pressure over the Aleutian Islands and Bering Sea, thus advecting moisture to NSA from the Gulf of Alaska. This cyclonic flow from the Gulf of Alaska is impeded by coastal mountain ranges at lower altitudes, but at higher altitudes moisture traverses the Alaskan peninsula and the Yukon and arrives at NSA mainly in easterly winds across the Beaufort Sea. Cluster 4 is the warmest and moistest cluster, and is distinct from Cluster 3 by relative westward displacement of the strong low pressure over the Aleutian and Bering Seas. This results in a direct and unimpeded path for warm and moist air through the Bering Strait to NSA. The multiyear time series of NSA data enable us to investigate the contrasting influences of these four clusters on the trends in radiative flux and MWR cloud properties during summer. Using ERA5 reanalysis data over the NSA region, every day between 1999-2024 can be identified with one of these four clusters.

Another important meteorological consideration involves atmospheric rivers (ARs). An AR is a narrow region of intense horizontal and vertically integrated water vapor transport (IVT) within a low-level jet most typically generated at the cold front of an extratropical cyclone (Ralph et al., 2018). ARs are now recognized as a major source of moisture transport





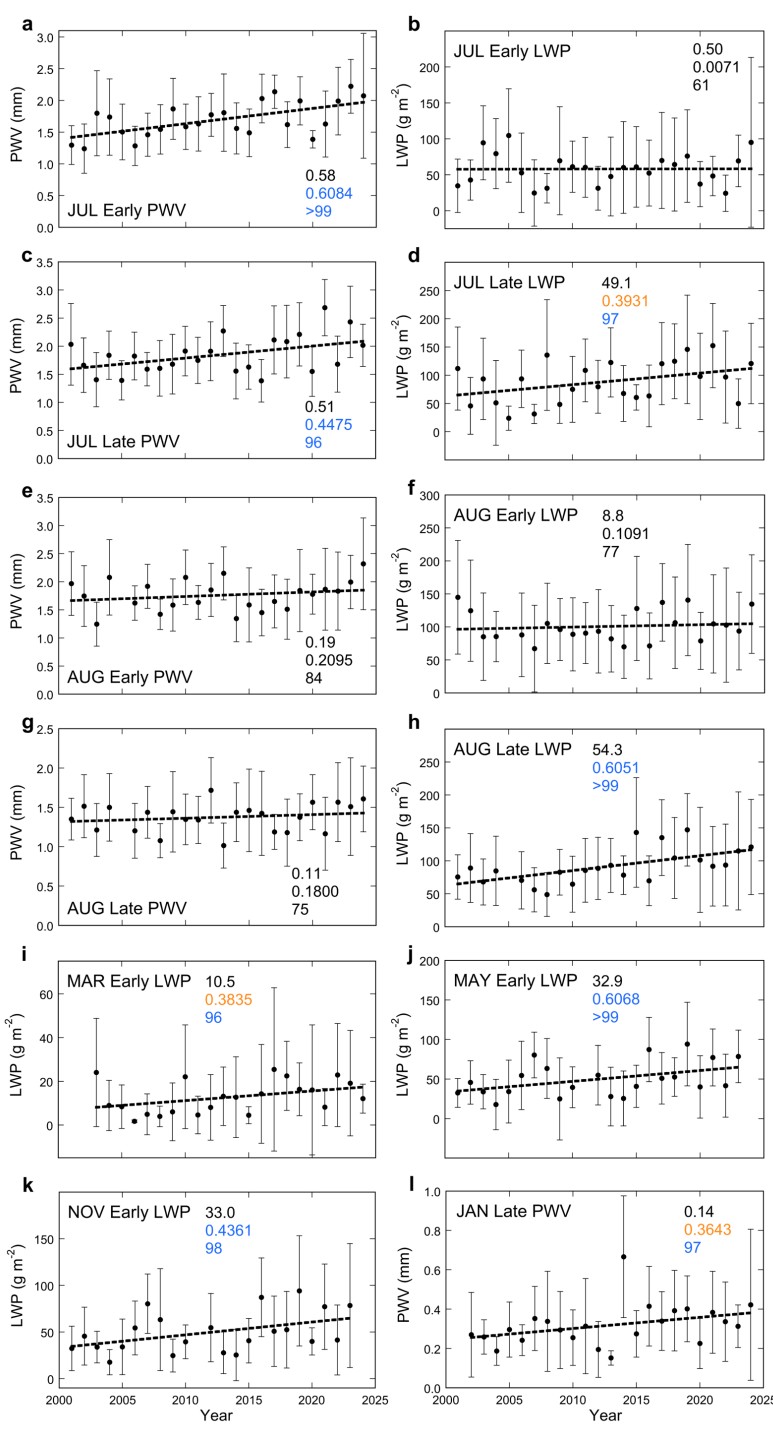

**Figure 4. Time series of the semi-monthly interval-average microwave radiometer-derived precipitable water vapor (PWV) and cloud liquid water path (LWP) for the four intervals in July and August (panels a-h), and for four other intervals that show some degree of statistical significance in the trend from 2001-2024 (panels i-l). Symbols and numbers on each panel are as in Figure 2.**



**Table 4. Summary of the trends in NSA MWR-retrieved precipitable water vapor and cloud liquid water path. *N* is the number of years having available data in each semi-monthly interval. Shown for each net radiative flux component are the linear change from 2001-2024, the Pearson correlation coefficient and the percent confidence in trend detection from a Mann-Kendall test. Two-tailed (one-tailed) significant correlations are indicated in bold (italic).**

| | | *PWV* (mm) | | | *LWP* (g m$^{-2}$) | | |
|---|---|---|---|---|---|---|---|
| Interval | N | $d_{01\text{-}24}$ | r | M-K (%) | $d_{01\text{-}24}$ | r | M-K (%) |
| JAN Early | 23 | -0.05 | -0.1078 | 65 | -0.8 | -0.0145 | 54 |
| JAN Late | 23 | *0.14* | *0.3643* | *97* | 8.0 | 0.3123 | 85 |
| FEB Early | 21 | 0.00 | -0.0033 | 51 | 6.3 | 0.2292 | 84 |
| FEB Late | 21 | 0.10 | 0.1883 | 76 | 3.8 | 0.1307 | 70 |
| MAR Early | 22 | 0.08 | 0.2212 | 84 | **10.5** | *0.3835* | **96** |
| MAR Late | 23 | 0.12 | 0.2665 | 79 | -1.5 | -0.0485 | 50 |
| APR Early | 24 | 0.15 | 0.1999 | 88 | 3.9 | 0.1537 | 51 |
| APR Late | 24 | 0.03 | 0.0631 | 50 | 3.0 | 0.0480 | 78 |
| MAY Early | 23 | 0.09 | 0.1963 | 79 | **32.9** | **0.6068** | **>99** |
| MAY Late | 23 | 0.05 | 0.0796 | 52 | 0.9 | 0.0165 | 70 |
| JUN Early | 24 | -0.16 | -0.2525 | 83 | 8.9 | 0.1290 | 65 |
| JUN Late | 24 | **0.25** | 0.2586 | **95** | 0.6 | 0.0069 | 68 |
| JUL Early | 24 | **0.58** | **0.6084** | **>99** | 0.5 | 0.0071 | 61 |
| JUL Late | 24 | **0.51** | **0.4475** | **96** | **49.1** | *0.3931* | **97** |
| AUG Early | 23 | 0.19 | 0.2095 | 84 | 8.8 | 0.1091 | 77 |
| AUG Late | 23 | 0.11 | 0.1800 | 75 | **54.3** | **0.6051** | **>99** |
| SEP Early | 24 | 0.09 | 0.1072 | 78 | -13.8 | -0.2131 | 78 |
| SEP Late | 24 | 0.24 | 0.3343 | 88 | 1.5 | 0.0165 | 59 |
| OCT Early | 23 | 0.03 | 0.0487 | 70 | -25.1 | -0.2616 | 89 |
| OCT Late | 21 | 0.16 | 0.3459 | 84 | -14.1 | -0.1134 | 63 |
| NOV Early | 22 | 0.14 | 0.3494 | 94 | **33.0** | **0.4361** | **98** |
| NOV Late | 22 | 0.03 | 0.0614 | 75 | 29.5 | *0.3888* | 90 |
| DEC Early | 22 | 0.04 | 0.0865 | 59 | 15.1 | 0.2947 | 79 |
| DEC Late | 22 | -0.02 | -0.770 | 50 | -0.4 | -0.0123 | 50 |





between midlatitudes and both northern and southern high latitudes (Newman et al., 2012; Wille et al., 2019; Mattingly et
al., 2020). The standard reference scale to characterize the impact of an AR involves a combination of IVT magnitude and
duration in a specific region as detected in satellite meteorological data, or weather forecast modelling including reanalysis
data (Ralph et al. 2019). This AR scale has been modified for polar regions (Zhang et al., 2024) and tested during the Year of
Polar Prediction (Bromwich et al., 2024). In polar regions the potential IVT range is smaller than over midlatitudes but
where the AR can nevertheless bring significant climatological impacts including extensive cloud cover and cryosphere
surface melt (Wille et al., 2019).

Between 1999-2024 we find using ERA5 an increasing trend in AR occurrence over the NSA site (Figure 5a). The JUL-Late
interval has been strongly influenced by ARs over NSA during the past two decades. The statistically significant trends in
SW, LW and Net fluxes, and PWV and LWP (Figure 2d-f, Figure 4c-d), nearly all lose statistical significance when we omit
the AR days (Figure 5b-f). This result does not apply to the other summertime intervals, whose radiative flux trends largely
remain significant when we omit the AR days (Table 4).

We examine the influence of individual meteorological regimes by comparing the trends evaluated for just one cluster with
those using all the data. The coldest Cluster 1 occurs infrequently during summer while the second coldest Cluster 2 occurs
most frequently (Figure 6a). We find a consistent influence of the warmer but relatively infrequent Cluster 3. During JUL-
Early the SW and LW flux trends which had modest significance (detected with MK but only one-tailed significance with
OLS) become more significant with OLS in only the Cluster 3 data (Figure 6b,c). Similarly in AUG-Early we see much
larger changes in SW, LW and Net fluxes in the Cluster 3 data compared with all the data, and under Cluster 3 a significant
upward LW trend is detected that does not appear using all the data (Figure 6g-i). In AUG-Late (Figure 6k,l) the upward
trend in LWP under Cluster 3 is more pronounced than in all the data, and the upward trend in PWV that is negligible in all
the data becomes statistically significant under just Cluster 3. We do not detect similar influences of the warmest and more
frequent Cluster 4. We do find a role for Cluster 2 during JUL-Late. Here the trends in SW, LW, and Net fluxes (Figure 6d-
f) and LWP (Figure 6j) all increase in statistical significance under just Cluster 2 compared with all the data. This is also the
interval strongly influenced by ARs, and Cluster 2 occurs 53% of the time, more frequently than in the other July and August
intervals.

## 4 Conclusions

After two and a half decades of data acquisition by the ARM NSA Facility the broadband radiometric data are beginning to
show statistically significant trends consistent with Arctic amplification. These are supported by emerging trends in MWR-
retrieved PWV and LWP. These results pertain mainly to summer but significant trends appear in some intervals in all
seasons. During summer the trends manifest as increasing LW net flux consistent with both a warming lower troposphere



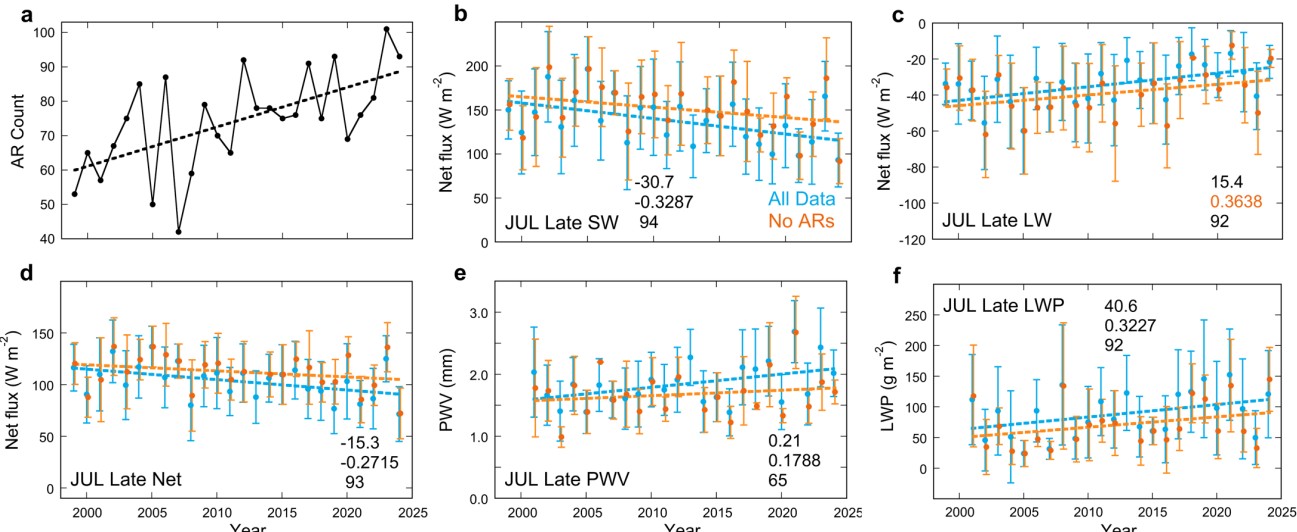

**Figure 5. Results involving the role of atmospheric rivers (ARs); (a) the total AR count over NSA in each year, with the linear trend shown as a dotted line; (b-f) time series for the Late July interval showing the net radiation flux components and MWR-measured quantities for all data in light blue and with the AR days removed in orange. Symbols and trend lines are as in Figure 2. The three numbers in each panel are also as in Figure 2, but are for the trends with the ARs removed.**

**Table 5. Summary of trends in the net radiative flux components and the MWR-retrieved quantities when all days containing atmospheric rivers are removed, presented as in Tables 2 and 3.**

| Net Fluxes (W m$^{-2}$) | | Shortwave | | | Longwave | | | Net Radiation | | |
|---|---|---|---|---|---|---|---|---|---|---|
| Interval | $N$ | $d_{99\text{-}24}$ | $r$ | M-K (%) | $d_{99\text{-}24}$ | $r$ | M-K (%) | $d_{99\text{-}24}$ | $r$ | M-K (%) |
| JUL Early | 26 | **-37.5** | **-0.4793** | **99** | **14.6** | **0.3917** | 94 | **-22.9** | **-0.4826** | **>99** |
| JUL Late | 25 | -30.7 | -0.3287 | 94 | *15.4* | *0.3638* | 92 | -15.3 | -0.2715 | 93 |
| AUG Early | 25 | **-24.8** | *-0.3764* | **95** | **11.4** | **0.3212** | **96** | **-13.4** | **-0.3608** | **95** |
| AUG Late | 26 | **-18.3** | **-0.4599** | **98** | 6.0 | 0.2338 | 94 | **-12.4** | **-0.4284** | **98** |
| *MWR* Retrievals | | *PWV* (mm ) | | | *LWP* (g m$^{-2}$) | | | | | |
| Interval | $N$ | $d_{01\text{-}24}$ | $r$ | M-K (%) | $d_{01\text{-}24}$ | $r$ | M-K (%) | | | |
| JUL Early | 24 | **0.61** | **0.5976** | **>99** | -6.2 | -0.1100 | 55 | | | |
| JUL Late | 23 | 0.21 | 0.1788 | 65 | 40.6 | 0.3227 | 92 | | | |
| AUG Early | 22 | 0.14 | 0.1365 | 77 | 14.1 | 0.1102 | 84 | | | |
| AUG Late | 24 | 0.15 | 0.2116 | 60 | **46.2** | **0.5435** | **98** | | | |








**Figure 6. Results involving the role of prevailing meteorological regimes identified by *k*-means clustering; (a) summary bar chart showing how the daily occurrences of ARs over the NSA site, summed over 1999-2024, are sorted into each of the four *k*-means clusters and each of the four bimonthly intervals for July and August; (b-l) time series of net radiative fluxes and microwave**

**radiometer-derived quantities for all data (black) and for only data from Cluster 2 (light blue) or Cluster 3 (green). Symbols and trend lines are as in Figure 2, but with the linear trend from the single-cluster data shown as a coloured solid line. The three numbers in each panel are the linear change between 1999-2024 (top), the Pearson correlation coefficient from ordinary least squares (middle) and the sample size for the single-cluster data (bottom).**





and increasing cloud LWP, with this increase being offset by a larger SW net flux decrease also due to increasing LWP. The result is a statistically significant decrease in Net surface flux throughout July and August between 1999-2024.

Although the summertime increases in PWV are consistent with a gradually warming lower troposphere, there is evidence that two synoptic-scale drivers play a larger role in these trends. The warm Cluster 3, associated with low pressure in the

Gulf of Alaska, exerts increasing influence on the LWP and surface radiative fluxes over NSA. In addition, the frequent low pressure over the Arctic Ocean (Cluster 2) is associated with increasing AR activity that also brings increasing PWV and LWP over NSA with concomitant trends in surface radiative fluxes. These effects may explain the relatively small surface warming trend at NSA during summer compared with other seasons and many other Arctic locations.

Satellite remote sensing studies covering earlier time periods (1982-2004) have shown radiative cooling effects in summer consistent with increasing cloud amount or optical depth (Wang and Key, 2005; Liu et al., 2008; Wang et al., 2012), but also report trends of different magnitudes and signs than in our more contemporary NSA observations. A recent satellite remote sensing study using NASA Clouds and Earth's Radiant Energy System (CERES) data from 2000-2020 projected that the ToE to 95% confidence in net surface radiative fluxes under all sky conditions for the Beaufort and Chukchi Seas are 26 and 22

years, respectively (Sledd and L'Ecuyer, 2021). Our summertime NSA trend detections after 26 years observed in situ are consistent with this projection. Another more recent multisensor satellite remote sensing study spanning 1996-2016 reports that over the Beaufort and Chukchi Seas during late spring (AMJ) the ToE to 95% confidence in surface cloud radiative forcing (CRF) are 29 and 24 years, respectively (Lelli et al., 2023). For their summer period (JAS) the same ToE is 24 years for both the Beaufort and Chukchi Seas. Also during their late spring (AMJ) over the Beaufort and Chukchi Seas they find

negative SW CRF trends and smaller positive LW CRF trends, for a negative Net CRF trend, which is qualitatively consistent with our results. Thus the NSA surface radiative flux measurements support these recent satellite remote sensing approaches.

Limitations with this study include the fact that the statistically significant trends are only starting to emerge. In all seasons

there are similar hints of trends that may be explained by Arctic amplification (Tables 2 and 3), but it remains uncertain whether a few more years of data will bolster or diminish the significance of these patterns. Nor is there any evidence that they will remain insignificant.

It is important to realize that the statistically significant results here pertain to only one location. They do not signify that the

entire Arctic will uniformly experience a surface radiative cooling effect in response to moisture advection from subarctic latitudes. For example, the Greenland Ice Sheet (GIS) is subject to very different synoptic-scale meteorology influenced by the North Atlantic Oscillation (NAO) (e.g., Ding et al., 2014; Pettersen et al., 2018) and the Icelandic Low (Serreze and Barry, 2014). Clustering analysis in that region conceptually similar to this work reveals several regimes that identify with



various phases of the NAO and El Niño Southern Oscillation in that North Atlantic – European region (Fereday et al., 2008). Over the GIS, optically thin clouds having LWP < 40 g m$^{-2}$ exert a unique Net radiative surface warming effect that can also inhibit refreeze of surface meltwater (Bennartz et al., 2013; Van Tricht et al., 2016). Other multidecadal surface radiation measurements in the high Arctic show contrasting trends from this work. Surface radiation measurements from western high Arctic land sites covering an earlier time period (1960s until 2004) show negative trends in downwelling SW flux that are qualitatively consistent with this work but that do not reach a threshold of statistical significance (Shi et al., 2010). Surface radiative flux measurements from Alert and Resolute Bay in the Canadian high Arctic, spanning 45-47 years up to 2004, show consistent positive annual net radiative flux trends and a correlation with the Arctic Oscillation (Weston et al., 2007). Baseline Surface Radiation Network (BSRN) measurements spanning 1992-2013 at Ny-Ålesund (Svalbard) show summertime increases in net radiation at the rate of ~8.4 W m$^{-2}$ per decade (Maturilli et al., 2015). More generally, the satellite remote sensing studies cited above show considerable spatial variability in surface radiative flux trends.

Nevertheless the results reported here are unique in showing the detection of statistically significant surface net radiative flux trends in direct measurements at a high Arctic coastal site, along with well characterized surface-based observations of trends in PWV and LWP that help explain the radiative flux trends in terms of the local atmospheric moisture content and its transport from both lower latitudes and the adjacent Arctic Ocean. These results highlight the value in establishing and maintaining well-equipped atmospheric observatories at remote high latitude sites capable of measuring the surface energy fluxes, tropospheric moisture and cloud properties, and explaining their atmospheric driving mechanisms.

**Data Availability**

All observational data are found in the US Department of Energy Atmospheric Radiation Measurement (ARM) Facility archive (www.arm.gov, and searchable by NSA site and measurement/instrument type). ERA5 data were obtained from the Copernicus Climate Data Store, doi:10.24381/cds.bd0915c6 and doi:10.24381/cds.adbb2d47.

**Author Contribution**

DL led the research effort including acquiring funding, organizing the project, performing NSA data analysis and drafting the manuscript. XZ provided the atmospheric river analysis and contributed to manuscript preparation. JM provided the *k*-means cluster analysis and contributed to manuscript preparation, AV contributed to funding acquisition, project organization and manuscript preparation. MC assisted with interpreting microwave radiometer data and contributed to manuscript preparation. DZ is the Instrument Mentor (ARM Facility database manager) for the NSA instruments used in this research, and contributed to manuscript preparation.



**Competing Interests**

The authors all affirm having no competing interests in this work.

**Financial Support**

This research was supported by the US Department of Energy under DE-SC0021974.

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
