# Peer review of "Surface Radiation Trends at North Slope of Alaska Influenced by Large-Scale Circulation and Atmospheric Rivers"

_EGUsphere, 2025_

## Author Response (AR2)

**Replies to Reviewers**

**Reviewer 1**

I thank the authors for their efforts in taking my comments into consideration. The manuscript reads well and is an important contribution to recent literature that capitalizes on long-term records at ground-based stations. I appreciate that the authors have included a new Section 3.3 on the role of natural variability in response to my major concern from my first review. Their method addresses that the root-mean-square-error (RMSE) in the net fluxes is greater than the measurement noise in their analysis, thus indicating that the measurement noise does not dominate the interannual variability and that the interannual trend is of meteorological origin. However, my intention was for the authors to address how natural variability obscures the long-term trend. I therefore recommend that the authors explicitly address this issue. E.g. how does the trend magnitude compare to the RMSE? Furthermore, I recommend that the authors include a statement about the role of natural variability on the trends based on this additional analysis in the Conclusions and Abstract of the manuscript.

*We are grateful for the reviewer's continued endorsement of this work, and we have included the requested statement in both the Conclusions and the Abstract of the revised manuscript.*

**Reviewer 2**

I'm happy with the revised manuscript. The authors have addressed all my questions and suggestions.

*These suggestions have substantially improved the paper.*